

# Microphysical properties and fall speed measurements of snow ice crystals using the Dual Ice Crystal Imager (D-ICI)

Thomas Kuhn[1] and Sandra Vázquez-Martín[1]

[1]Luleå University of Technology, Department of Computer Science, Electrical and Space Engineering, Division of Space Technology, 98128 Kiruna, Sweden

**Correspondence:** Thomas Kuhn (thomas.kuhn@ltu.se)

**Abstract.** Accurate predictions of snowfall require good knowledge of the microphysical properties of the snow ice crystals and particles. Shape is an important parameter as it influences strongly the scattering properties of the ice particles, and thus their response to remote sensing techniques such as radar measurements. The fall speed of ice particles is another important parameter for both numerical forecast models as well as representation of ice clouds and snow in climate models, as it is

responsible for the rate of removal of ice from these models.

We describe a new ground-based in-situ instrument, the Dual Ice Crystal Imager (D-ICI), to determine snow ice crystal properties and fall speed simultaneously. The instrument takes two high-resolution pictures of the same falling ice particle from two different viewing directions. Both cameras use a microscope-like set-up resulting in an image pixel resolution of approximately 4 µm/pixel. One viewing direction is horizontal and is used to determine fall speed by means of a double

exposure. For this purpose, two bright flashes of a light emitting diode behind the camera illuminate the falling ice particle and create this double exposure and the vertical displacement of the particle provides its fall speed. The other viewing direction is close to vertical and is used to provide size and shape information from single-exposure images. This viewing geometry is chosen instead of a horizontal one because shape and size of ice particles as viewed in the vertical direction are more relevant than these properties viewed horizontally as the vertical fall speed is more strongly influenced by the vertically viewed

properties. In addition, a comparison with remote sensing instruments that mostly have a vertical or close to vertical viewing geometry is favoured when the particle properties are measured in the same direction.

The instrument has been tested in Kiruna, northern Sweden (67.8°N, 20.4°E). Measurements are demonstrated with images from different snow events, and the determined snow ice crystal properties are presented.

# 1 Introduction

Accurate knowledge of atmospheric ice crystals and snowflakes, or snow particles is needed for meteorological forecast, climate and forecast models (see, e.g., Tao et al. (2003); Stoelinga et al. (2003)). In particular, microphysical properties that



are difficult to measure with remotes sensing such as size, area, shape, and fall speed are important. This knowledge can, for instance, help improve parameterizations of snow particles in atmospheric models.

To retrieve snow depth or precipitation amount reaching the ground, the microphysical properties of the snow particles on their way down have to be known. Fall velocity plays a significant for modelling of the microphysical processes. It determines the number of particles present in the measuring volume and the snowfall rate, or the rate of their removal from clouds.

Other interesting and important microphysical properties of snow particles are their shape and morphology, not only for investigating growth processes. Snow particle shape and morphology affect radar measurements (Sun et al., 2011) and microwave

brightness temperature (Kneifel et al., 2010), and are significant for optical remote sensing retrievals of cloud properties (see, e.g., Yang et al., 2008; Baum et al., 2011; Xie et al., 2011; Loeb et al., 2018).

Snow fall has long been monitored by ground-based instruments. However, instruments that can measure size, shape, and fall speed at the same time are still scarce. Instruments can be classified into different categories depending on what is measured primarily. Disdrometers, for example, measure shadow or attenuation as droplets or snow particles pass one or several light

beams. Fall speed can be measured or estimated by these disdrometers from time between the two beam passages in case of two parallel beams with known vertical spacing or from the duration of attenuation. Three common examples are Parsivel (Particle Size Velocity disdrometer, see, e.g., Battaglia et al., 2010), 2-DVD (Two-Dimensional Video Disdrometer, see, e.g., Kruger and Krajewski, 2002) and HVSD (Hydrometeor Velocity and Shape Detector, see, e.g., Barthazy et al., 2004). The latter two are optical array instruments, where the shadow is detected with a linear array of detectors. Thus, a shadow image

can be reconstructed and shape discerned. These disdrometers, however, are designed for larger snowflakes and have lower size limits (or pixel sizes) of around $200\,\mu m$.

Another category of instruments uses camera systems for optical imaging of snow particles. One example is SVI (Snowflake Video Imager, in a newer version also called PIP, Particle Imaging Package), a video camera and halogen lamp placed at about $2\,m$ distance for background illumination. It has a pixel resolution of $100\,\mu m$. The higher frame rate ($380\,s^{-1}$) of PIP allows

to determine the fall speed with image analysis software that follows particles over several frames. The Ice Crystal Imaging probe (ICI) uses a high-resolution CCD camera system with a pixel resolution of $4.2\,\mu m$ (Kuhn and Gultepe, 2016). It has also been used to measure fall speed by double-exposing snow particles using two flashes of illuminating light issued at a know time difference.

There are instruments designed for aircraft that have also been used on the ground for snow measurements. CIP (cloud

imaging probe, see Baumgardner et al., 2001) is an optical array probe and has been used on ground as GCIP (Gultepe et al., 2014). VIPS is a video camera system (see Appendix of McFarquhar and Heymsfield, 1996) with a high pixel resolution. On ground it has been used for example for ice fog particles with a pixel resolution of $1.1\,\mu m$ (Schmitt et al., 2013). CPI (Cloud Particle Imager) uses a CCD camera to produce shadow graphs, or images if the ice particle is in focus, with a pixel resolution of $2.3\,\mu m$. All three instruments used aspiration to produce similar inlet flows as encountered on the aircraft.

Holographic imaging has the advantage of a larger depth of field when compared to depth of field for 'in-focus imaging'. Shadow-like images of out-of focus particles can be reconstructed and their position determined. Holographic Detector for Clouds (HOLODEC) is an aircraft instrument (Fugal et al., 2004) and HOLIMO (Holographic Imager) is a ground-based



instrument (Amsler et al., 2009). HOLIMO II, a newer version, is used for ground-based in-situ measurements of particles in mixed-phase clouds (Henneberger et al., 2013).

PHIPS (Particle Habit Imaging and Polar Scattering) uses a combination of optical imaging and scattering (with polar nephelometer). A first version of the instrument had a high pixel resolution, better than the 2 µm optical resolving power (Schön et al., 2011). The next version, PHIPS-AIDA, added a second camera at an angle of 60° to the first camera to produce stereo images and has been used for cloud chamber experiments (Abdelmonem et al., 2011). MASC (Multi Angle Snowflake Camera) uses three cameras to image snow from three angles, while simultaneously measuring their fall speed with two sets

of IR emitter–receiver pairs registering the shadow twice (Garrett et al., 2012). The cameras are viewing horizontally and are separated by 36°. Different pixel resolutions may be used by the cameras, and the version described by Garrett et al. (2012) used pixel resolutions between 9 and 32 µm. Such multi imagers provide more detail about the 3D structure of the snow particle that adds valuable information to the microphysical data collected by imaging instruments. This is useful, for example, to provide better estimates of snow particle mass.

Here, another multi imager, the Dual Ice Crystal Imager (D-ICI) is presented that uses two cameras for simultaneous imaging and fall speed measurements. It is a further development of ICI that adds a second camera. While the first camera is using a horizontal viewing direction, the second camera is viewing the falling snow particle vertically. As already described by Kuhn and Gultepe (2016) for ICI, the side-view camera can produce double-exposures for fall-speed measurements. The second camera, providing top-view images, which are more relevant when comparing to fall speed. The cross-sectional area

as seen from the top is better related to the particle drag and terminal fall velocity than area determined from side-view. Additionally, particle size and area from top view are also more useful when comparing to optical remote sensing, which often uses vertical viewing geometries too. Sections 2 and 3 describe the design of D-ICI and data processing methods, Sect. 4 presents measurements to evaluate the instrument's capabilities.

## 2    Instrument

### 2.1    Instrument set-up

D-ICI uses passive sampling with a vertically pointing inlet. Its set-up can be seen schematically in Fig. 1. Ice particles, small ice crystals, snow crystals, and snowflakes, falling into the inlet will fall down the sampling tube and traverse the optical cell. In the centre of the optical cell is the sensing volume. If a particle is falling through the sensing volume it is detected by the detecting optics. Upon detection, the ice particle is optically imaged from two different directions. Figure 2 shows an example

of the resulting pair of images for one ice particle. One of the two viewing directions is looking horizontally from the side, called side view, and the other vertically from the top, called top view. The former will allow to measure the fall speed, if using a double-exposure (see Sect. 3.3). The latter will provide area and shape as seen in vertical direction, which are more relevant for fall speed and radiative properties. Because particles fall vertically, an exact vertical viewing geometry for the top view is difficult to achieve as part of the optics would obstruct the particles' motion. Therefore, the top view is a near-vertically

viewing configuration that looks through the optical cell inside the vertical sampling tube at a shallow angle to the vertical axis



(17 °). The side view, on the other hand, uses exactly a horizontal viewing geometry. Figure 3 shows a photograph of D-ICI, and a more detailed description is given in the following sections.

## 2.2 Inlet and sampling tube

Similarly to the ICI probe (Kuhn and Gultepe, 2016), also D-ICI has a funnel-shaped inlet, wider at the top, with a sharp upper
edge (see Figures 1 and 3). Ice particles fall freely into this inlet. The inlet has a diameter of 25 mm at the top and narrows down
to an inner diameter of 8 mm. It is concentrically mounted atop of the vertical sampling tube with inner diameter of 12 mm.
Ice particles falling through the inlet are therefore transferred into the sampling tube. After falling about 160 mm vertically
through the sampling tube, ice particles come to the section containing the sensing volume. The length of the sampling tube
upstream of the sensing volume is sufficient (more than ten times the diameter of the sampling tube) so that particles can relax
from any effects of wind. Hence, the fall speed of ice particles is not affected by wind or turbulence.

## 2.3 Imaging optics

In the sensing volume (see Sect. 2.4), particles are optically imaged by two imaging systems, each using a a monochromatic
CCD camera (Chameleon 1.3 MP Mono USB 2.0, Point Grey, now FLIR) having a $1280 \times 960$ pixel sensor chip with pixels that
are 3.75 μm $\times$ 3.75 μm in size. These camera systems are similar to the microscope optics used in ice crystal imaging set-ups
with single imaging systems (Kuhn et al., 2012; Kuhn and Gultepe, 2016). They consist of a microscope objective followed by
a tube lens, as indicated in Fig. 1. For the horizontal view, i.e. side view, the microscope objective (RMS4X, Thorlabs) has a
focal length of 45 mm. For the top-view system, the objective is a single convex lens, a positive achromatic doublet (AC254-
050-A, Thorlabs) with focal length of 50 mm. This has, compared to the microscope objective, a longer working distance of
43 mm, which is required for the top-view configuration.
110       The tube lens of the side-view optics is a positive achromatic doublet (AC254-045-A, Thorlabs) with the same focal length
as its microscope objective, 45 mm. As tube lens of the top-view optics, the same achromatic doublet as for its objective is
used. Thus, the resulting magnifications are the same for both systems, $M = 1$. Both camera systems have therefore a pixel
resolution, i.e. the size of a feature of the imaged object that appears on the image as one pixel, equal to the pixel size of
3.75 μm. And the field of view (FOV) is equal to the exposed sensor area, i.e. 4.8 mm $\times$ 3.6 mm.
115       Both imaging systems use bright-field illumination from the back. This is achieved by a light emitting diode (LED) with
a simple focusing lens optics allowing for an even illumination of the FOV. Each of these two lens–LED configurations is
arranged along the optical axis of the respective imaging optics on the opposite side of the sensing volume (see Fig. 1).
     The top-view optical system uses a mirror between the sensing volume and the objective lens. This allows to fold its optical
axis so that it is parallel to the optical axis of the side-view system for a simpler mechanical set-up.



## 2.4 Detection and sensing volume

The sensing volume, i.e. the volume in which particles are detected and imaged, is defined as the intersection of the laser beam for detection with the overlapping FOVs of the imaging systems. The laser beam, which has a wavelength of 650 nm and power of 1 mW, is aligned perpendicular to the optical axes of both imaging optics. It is shaped by an aperture to about 1 mm horizontal width, which corresponds approximately to the depth of focus of the side-view camera. The laser beam is further shaped by a cylindrical lens (LJ1960L1, Thorlabs) with focal length of 20 mm such that its vertical height, originally about 3 mm, is focused to approximately 0.1 mm in the centre of the FOV of the side-view camera. Thus, the laser beam forms a light sheet with width of approximately 1 mm and height of 0.1 mm. Both the side- and top-view cameras are focused so that their focal planes are aligned with this resulting laser sheet. As a consequence, all detected particles are in focus for both images.

To determine the snowfall rate or the snow crystal number concentration, the sensing area, i.e. the area through which detected particles fall, needs to be known rather than the sensing volume. The sensing area is the horizontal cross section of the sensing volume (i.e. the cross section perpendicular to the vertical falling motion). The area is therefore given by the product of the width of the FOV of the cameras and the sum of laser beam width (1 mm) and particle size. This sum has to be used instead of laser beam width only, because particles that are only partially in the laser beam will be detected too. Thus, the sensing area is size dependent (larger particles have a larger sensing area). When assuming a constant sensing area corresponding to a particle size of 500 μm, the concentrations of particles larger than this size would be overestimated. This overestimation is compensated by the size-dependent probability of a particle to touch one of the image borders. Larger particles are more likely to touch an image border, i.e. to be partially outside the image. Such ice particles that touch one image border are therefore excluded from data analysis (see Sect. 3.2). This exclusion from further analysis results in an underestimation of larger particles, hence compensating the overestimation due to size-dependent sensing area. Thus, the assumption of a constant sensing area does not cause a significant uncertainty, as was also discussed by Kuhn and Gultepe (2016), and the sensing area to be used is 4 mm × (1 mm + 500 μm) = 6 mm$^2$. Here, we use 4 mm as FOV instead of 4.8 mm mentioned earlier due to the fact that the FOV of the top-view camera is somewhat restricted as a consequence of incomplete illumination of the whole camera FOV (see Sect. 3.2 and Fig. 4 for an example of a complete image).

Scattered light from the part of the laser beam within the sensing volume is collected and focused on a photodiode (FDS010, Thorlabs) by two plano-convex detector lenses (LA1951-A, Thorlabs). The photodiode is located along the axis of the laser beam, which is stopped by a light trap mounted in the centre of the first lens. The diameters of the light trap and the lens tube holding the detector lenses are such that the photodiode detects light scattered by ice particles in the sensing volume in near-forward direction in the range of scattering angles between approximately 10° and 32°. The photodiode has a circular sensitive area with a diameter of 1 mm. Its small area means that most particles that are outside the sensing volume, but still in the laser beam, do not scatter light that can be detected by the photodiode. This minimizes false triggers, i.e. detected scattering leading to empty images as particles are outside the FOVs of the two cameras.

The current of the photodiode is converted to a voltage and amplified (effective current-to-voltage amplification of 2.2 MΩ). The resulting photo-detector voltage, proportional to the scattered light's intensity, is compared to a threshold voltage (approx-





imately 0.15 V). A trigger signal is issued whenever the photo-detector voltage is larger than this threshold. The trigger signal

is used to trigger the two images to be taken of the detected ice crystal as well as the two background-illuminating LED flashes. Hence, all particles larger than a certain threshold size are detected and then imaged. With the help of Mie scattering calculations (see, e.g., Bohren and Huffman, 1983) this threshold size (diameter of spherical ice) can be estimated as approximately 10 µm.

### 2.5  Computer and data collection

Both imaging systems are triggered by the same signal (see Sect. 2.4). To guarantee simultaneous imaging by the two cameras, each of the two imaging systems has its own dedicated computer for operation and data collection. In this way, there are no particular requirements about the computer's performance, and two Raspberry Pi's are used for D-ICI. Each computer stores its own image data on an SD card, which is connected to the computer's USB port via a card reader. One of the two computers acquires also temperature inside and outside the instrument, registered by two thermistors, and the outside relative humidity

with a HIH-4000 sensor (Honeywell) with an accuracy of $\pm 3.5\%$.

Both computers are connected to a network via ethernet cables. This allows to synchronize them with each other. Consequently, corresponding side- and top-view images can be recognized by their time stamp, which is part of the file name. Both computers can be accessed through an additional laboratory or office computer, which is connected to the same network via cable or internet, if the network provides internet access. Data can then be retrieved using this laboratory computer. Alterna-

tively, the SD cards can be collected to copy the image data. Then, the image data will be processed by the laboratory computer as described in Sect. 3.2.

## 3  Methods

### 3.1  Snowfall rate and number concentration

While the focus of D-ICI is high-resolution images for shape and fall speed measurements, snowfall rate and number concen-

tration can also be determined from the acquired data. For that, here snowfall rate $r_s$ is defined as number of snow crystals falling on a given area during a given sampling time $t$. The inlet is sampling falling snow crystals from a larger area than the cross section of the sampling tube, which results in an enhanced number of snow crystals in the sampling tube. To account for this enhancement, an effective sensing area $A$ is used. It is larger than the sensing area by a factor equal to the ratio of the areas of the 25-mm inlet and the 12-mm sampling tube, i.e. a factor of 4.3. This yields $A = 4.3 \cdot 6\,\text{mm}^2 = 26\,\text{mm}^2$. Then, $r_s$ is

determined as number of snow crystals $N$ divided by the effective sensing area $A$ and sampling time $t$:

$$r_s = \frac{N}{At}. \tag{1}$$

The number concentration $n$ is calculated from $N$ divided by the sampling volume $V$. To determine $V$ an average fall speed $v$ of $0.5\,\text{m}\,\text{s}^{-1}$ is assumed. With this assumption, the effective sampling flow rate of D-ICI becomes $Av = 13\,\text{cm}^3\,\text{s}^{-1}$. Finally, $n$



is calculated using Eq. 2.

$$n = \frac{N}{V} = \frac{N}{Avt} \tag{2}$$

While the uncertainty due to size dependencies cancels out to a good approximation (see Sect.2.4), the assumption of constant snow fall speed $v$ introduces a new uncertainty. An additional uncertainty is related to the estimation of the effective sensing area. This affects both $n$ and $r_s$. Hence, $n$ and $r_s$ determined with D-ICI should be considered estimates of the actual number concentration and snowfall rate.

## 3.2 Image processing

The images have pixels with grey levels between 0 (black) and 255 (white). An automated image processing algorithm is applied to all top-view images to retrieve ice particle size, area, area ratio, and aspect ratio. It first removes non-particle features from the background. Then the particles on the images are detected and their edges are found. This algorithm has been used by Kuhn and Gultepe (2016); Vázquez-Martín et al. (2019) and is a simplified implementation in Matlab of the algorithm described in Kuhn et al. (2012). Here, we summarize this implementation briefly. In the following the different steps of the algorithm are described, of which some are shown in Fig. 4 for an example image.

A background image without any ice particle is used to correct for uneven background illumination, i.e. remove non-particle features from the background. For this, the difference between background and image to be analysed is determined. The difference is positive where the presence of a particle makes the image darker than the background. For regions where the image is brighter than the background, the resulting negative values are set to zero. These are usually only regions within an ice particle, where transmitted light can appear as brighter spot, surrounded by darker features or the edge of the particle. Now, images are rejected from further analysis if no particle was captured on them, i.e. images that are very similar to the background. For this, a lower threshold is applied to the difference. The image is rejected if the difference does not exceed the threshold for any pixel. A suitable threshold is 30; images with ice particles exceed this by a large margin.

Then, for the remaining images, the difference to the background is first scaled to increase the dynamic range of grey values. This is done for each pixel individually, so that the possible maximum difference (when image pixel is black), at any background pixel becomes 255. Effectively, the scaling factor at any pixel is $255/\mathrm{bg}$, where $\mathrm{bg}$ is the grey level of the corresponding background pixel. To avoid large scaling factors where the background is dark ($\mathrm{bg}$ is small), the factor is limited to 2.5. For very dark background ($\mathrm{bg} < 20$) the scaling is set to 1. This scaled difference is then inverted by subtracting it from 255, so that the resulting grey level image represents the image cleaned from background features. This can be seen for an example image in Fig. 4, where panel a) shows the original image and b) the image after the background has been removed. Regions of the original image that were identical to the background or had brighter spots are now white (255) in this cleaned image, and regions where the original image was darker than the background show now grey levels ($< 255$).

The following steps in the image processing apply to the cleaned image resulting from the background removal described above. For detecting in-focus particles two thresholds are applied, a grey-level threshold and a gradient threshold. The grey-level threshold is used to find particles and their edges, and the gradient threshold is used to reject out-of-focus particles. First,





images that do not have any pixel darker than the grey-level threshold are discarded. This rejects particles that are much out of focus. Then, a binary mask, i.e. a black-and-white image, of the same dimension as the original image is created where logically True entries represent image pixels that are darker than the grey-level threshold. The binary mask is then smoothed to

remove variations at the one-pixel level, which are considered to not reflect the actual variations in the edge of the ice particle. The smoothing is achieved by first dilating each True pixel in the binary mask so that the four neighbouring pixels (above, below, right, and left) will also be True. Then, the dilated binary mask is eroded, to restore its original size, by setting the four neighbours of each False pixel to be also False. Between the dilation and erosion steps, the binary mask is also filled, i.e. all pixels that are False but completely enclosed by True pixels are converted to True. This will include the brighter spots,

which many ice crystals show on the images, to the particle they belong to. Then, on the resulting black-and-white image (see example in Fig. 4 c), ice particles are represented by connected True pixels in the binary mask.

All particles, i.e. regions of connected pixels that are now included in this binary mask are then identified and their edges are found (with the Matlab function bwboundaries). For each particle, this results in both a list of coordinates of the edge pixels and a mask containing all pixels that belong to the particle. Each particle can then be processed individually.

Firstly, out-of-focus particles are rejected. For this purpose, a gradient matrix is computed from the image. The values of this matrix, are used as a parameter indicating in- or out-of-focus particles. For computing the gradient values, the image is filtered (using the Matlab function imfilter) with a Sobel horizontal edge-emphasizing filter (generated with the Matlab command fspecial('sobel')) and its transpose, i.e. with the corresponding vertical filter. The resulting matrices represent the horizontal and vertical gradients, respectively. The values of the gradient parameter are then calculated as the sum of the absolute values

of these horizontal and vertical gradients (Kuhn et al., 2012). For each particle, the maximum gradient value of all pixels associated to that particle is then compared to the gradient threshold. The particle is rejected as out-of-focus if this maximum is lower than the threshold. For the example image of Fig. 4, two ice particles are found using the grey level threshold (see panel c), however, one of these two particles is rejected based on the low values in the gradient matrix shown in panel d).

Secondly, particles with apparent problems are marked with quality flags. A particle that is in part out of focus can sometimes

have parts of the edge not being detected yielding an apparently fragmented edge with narrow gaps. Similarly, if thin ice particle features result in brighter pixels than the grey-level threshold, a fragmented edge is the consequence. To account for this, two or more detected particles that appear very close to each other are joined and the resulting particle is marked as being 'fragmented'. The area of such a particle as determined from the detected pixels will be too small. The resulting error is not large, because the gaps are only small, and by joining the fragmented pieces, the particle may still be considered. However,

being marked, it can also easily be excluded from further analysis. An example of an ice particle detected with fragmented edge is given in Fig. 5, panel b). The other ice particle in the same figure shows the un-fragmented edge of the example particle from Fig. 4. In addition, particles that are touching the image border are marked with another flag as 'on-border'. Their size and area are under-estimated as they are in part outside the image. Thus, using this flag they can be excluded from analysis when size and area matter. Figure 6 shows an example of an ice particle with the 'on-border' flag. A further problem is related

to incomplete illumination of the top-view images due to restricted geometry in the longer light path in top-view compared to side-view optics. This results in dark corners where ice particles cannot be seen. Consequently, also particles touching these





dark corners have to be excluded from analysis as their size cannot be know, similarly as for 'on-border' particles. To allow this, these particles are marked with an additional flag as 'in-darkregion' when they have at least one pixel within the dark corners. For this, a mask containing the corresponding dark pixels (darker than a certain threshold) in the corner regions is
constructed from the background image. Figure 6 shows an example of an ice particle with the 'in-darkregion' flag.

Lastly, area and size information is determined for each detected ice particle. As parameter describing a characteristic size of the detected particle we are using maximum dimension, i.e. the diameter of the smallest circle that completely encloses that particle on the image (see Fig. 5 for an example). The area corresponds to the number of pixels that represent the particle in the binary mask. Both size and area are given in units of pixels. They are then converted to actual length and area by multiplying
with the pixel resolution and squared pixel resolution, respectively.

As this method is the same as used for the imager described by Kuhn et al. (2012), which used similar optics, sizing accuracy is expected to be similar. There, the determined size of a small particle (about 50 pixel in size) varied by about two pixels when the location within the depth of focus was changed. Larger inaccuracy is avoided by rejecting out-of-focus particles. To this uncertainty, one pixel should be added to account for uncertainty of particle edge location. Thus, a combined sizing accuracy
of approximately $10\,\mu m$ (or three pixels) is expected for D-ICI. Consequently, for a $200\,\mu m$ particle, the expected error in area should be on the order of 10%.

### 3.3  Snow fall speed measurement

The side-view camera can be operated in a fall-speed mode, in which the falling ice particle is captured twice on the same image by using a double exposure. This concept has been tested with ICI in a configuration without inlet, so that ice particles
could freely fall through the instrument (Kuhn and Gultepe, 2016). For D-ICI, the inlet and sampling tube are designed so that particles fall vertically undisturbed before they reach the sensing volume, thus the set-up does not need to be modified to allow measurements of fall speed. In the fall-speed mode, two very short illumination flashes are used, which have a time separation of $\Delta t = 1.26\,\mathrm{ms} \pm 0.01\,\mathrm{ms}$. This time difference is long enough to yield a clear separation of the two particle appearances on the image, but also short enough so that the particle does not fall out of the vertical FOV of the imaging optics. Thus, the
particle's fall speed $v$ can be determined from the vertical fall distance $\Delta s$, as measured on the image, and the time separation $\Delta t$ of the two exposure flashes simply as

$$v = \frac{\Delta s}{\Delta t}. \tag{3}$$

The vertical fall distance $\Delta s$ is measured by manual inspection of the side-view images. Two or three points at extremes of each particle to be analysed (e.g. a far right corner and far left corner point) are identified and their coordinates on the image
are recorded. The same points are then also identified and recorded on the second appearance of the particle on the image, and the vertical distance is determined as the difference of the vertical coordinates of pairs of corresponding points of the two appearances. From the two or three vertical distances determined in this way an average vertical fall distance is calculated.

While falling, the difference of the horizontal coordinates is usually close to zero. Such a difference could be caused by sideway or rotating (tumbling) motion. Horizontal winds, which affect other instruments, with an open sampling volume, such





as PIP and MASC do not cause a sideway motion in the enclosed sensing volume of D-ICI. Thus, only a tumbling particle can
be responsible for a difference of the horizontal coordinates, and tumbling of ice particles is not often seen (see Sect. 4.2). If it
occurs, it is detected by significantly different values of the individual vertical distances measured for a point on the right and
left side of the particle, respectively, so that particles that are tumbling too much may be excluded from analysis of fall speed
data. When tumbling, one side of the snow particle falls faster and one slower than the average that is determined from the

averaged fall distances $\Delta s$. Thus, by rejecting tumbling particles, e.g. those that rotate by more than $10°$, the error in fall speed
can be limited to approximately 7%. Uncertainties in $\Delta t$ have a negligible effect on fall speed error. Also the error related to
accuracy of point selection (about two pixels), which translates to an additional uncertainty in $\Delta s$, however, only on the order
of 1%.

While side-view images are not processed automatically, the top-view images are (see Sect. 3.2). Results from this automatic

processing of top-view images provide size, area, area ratio, and aspect ratio for the particles, whose fall speeds are determined
from the corresponding side-view images.

## 4   Measurements

### 4.1   Images and shapes

According to the design, the pixel resolution should be equal to the pixel size of the CCD cameras, $3.75\,\mu\text{m}$ (see Sect. 2.3).

This has been confirmed by imaging a calibration target, a graticule with $10\,\mu\text{m}$/division and total length of 1 mm. The lengths
in pixels corresponding to 1 mm from several such images have been converted to pixel resolutions yielding an average of
$3.74\,\mu\text{m}$/pixel with a standard deviation of $0.02\,\mu\text{m}$/pixel for the side-view imaging optics, and, from separate images, the same
values for the top view.

Figure 7 shows a few examples of ice particle images from snow fall in early winter (2014-10-23 in Kiruna), when ambient

surface temperature was about $-5\,°\text{C}$. Each ice particle is shown in the two views, where the top view is shown in the upper
panel and the corresponding side view in the lower panel. These detailed images of ice particles allow to recognize their shapes.
On 2014-10-23 the ice particles had predominantly bullet-rosette and similar shapes, but also plate-like and capped-column
shapes (see Fig. 7). On another day, 2014-10-19, with similar ambient surface temperatures of about $-3$ to $-6\,°\text{C}$, two dominant
shapes were observed, graupel (heavily rimed snow crystals) and rimed needles (see Fig. 8). Most of the rimed needles on that

day seemed to be agglomerates or ensembles of two or more single needles (called bundles of needles by Magono and Lee,
1966).

### 4.2   Fall speed

Figure 9 shows examples of double-exposed images from the side view, showing the falling ice particles twice, used to deter-
mine fall speed. The images from 2014-10-19 (top row of Fig. 9) also include a few drizzle droplets. The heavy riming on that

day indicates the presence of cloud droplets, and the imaged drizzle droplets originate from such cloud or fog droplets that





have grown large enough to precipitate and fall into the inlet of D-ICI. They were, with only very few exceptions, smaller than all snow particles.

One of the particles shown in Fig. 9 is tumbling (right-most ice particle in lower row). The rotation, around an axis perpendicular to the image plane, of the particle between the two exposures is approximately 8°, which seems still acceptable if one wants to determine fall speed with an error of below about 10%. Hence, 10°, or perhaps up to 15°, may be used as limit, above which the image has to be discarded for fall speed measurement. Selecting a few days randomly and analysing the ice particle images on those days (total of 946 particle images) yields that approximately 8% of ice particles are tumbling by more than an angle of 10°, and only 3% more than 15°. This means, that particles in general tumble somewhat, however, the majority of ice particles tumbles so little in the time between the two side-view exposures, that fall speed can still be measured.

### 4.3 Cross-sectional area

Using the top-view images, the ice particles' projected area in fall direction (i.e. area projected on a surface perpendicular to the vertical fall direction) can be determined. Figure 10 shows these projected, or cross-sectional areas $A$ from snowfall measured on 2014-10-19 between approximately 6 and 19 UTC (at temperatures on the ground between $-3\,°\mathrm{C}$ and $-6\,°\mathrm{C}$) as a function of particle size, i.e. maximum dimension $D$ also determined from the top-view images. On this logarithmic plot, the cross-sectional area of spheres having a diameter equal to the maximum dimension is represented by a straight line given by $A = \pi/4 \cdot D^2$. A power law $A = \gamma D^\beta$ can be fitted to the data to find the parameters $\gamma$ and $\beta$. For the data shown in Fig. 10 this yields $A = 4.72 \cdot 10^{-11}\,\mathrm{m}^2 \cdot D^{1.24}$, $D$ in µm, with a correlation coefficient $R^2 = 0.71$.

When the ice particles are classified according to their shapes, power laws can be fitted to the resulting subsets of data to find relationships describing area for specific shapes. On 2014-10-19 two dominant shapes were observed, graupel and rimed needles (see Fig. 8). The fitted power laws for these two shapes are indicated in Fig. 10 by coloured lines and are given by

$$\mathrm{Graupel:} \quad A = 7.89 \cdot 10^{-13}\,\mathrm{m}^2 \cdot D^{1.93}, \quad (R^2 = 0.97) \tag{4}$$

$$\mathrm{Rimed\ needles:} \quad A = 1.63 \cdot 10^{-12}\,\mathrm{m}^2 \cdot D^{1.64}, \quad (R^2 = 0.78) \tag{5}$$

with $D$ in µm. The groups of particles used for these fits are shown in Fig. 10 as coloured symbols and correspond to a selection of the most compact-looking graupel and almost all particles that could be identified as rimed needles.

The images from 2014-10-19 also show a few drizzle droplets, which can be seen in Fig. 10 with areas very close to the area–dimensional relationship for spheres. Droplets are the smallest particles measured by D-ICI on that day, with maximum dimensions of below 200 µm for the smallest droplets. Due to their spherical shapes, the determined area ratios were very close to 1, and all particles with area ratio larger than 0.9 were droplets. For these, the fitted area–dimensional power law is $A = 6.79 \cdot 10^{-13}\,\mathrm{m}^2 \cdot D^{2.02}$ ($D$ in µm, $R^2 = 1.00$), which is very close to the cross-sectional area of spheres.

When looking at the area–dimensional relationship for a certain shape, the fit to the power law can be very good. An exception here are rimed needles. However, they still have a fairly good fit, better than the fit to all data with one common power law, which would predict poorly the area for any of the shapes here, droplets, graupel, and rimed needles (see Fig. 10). Figure 10 also shows for comparison two relationships reported by Mitchell (1996), one for rimed long columns (as thin line





in magenta) and one for lump graupel (blue). While the latter agree very well with our graupel, the rimed long columns have a

larger cross-sectional area than our rimed needles, which one would expect if columns are compared to thinner needles.

### 4.4   Fall speed measurements

Fig. 11 shows the fall speeds versus the maximum dimension of individual ice particles from the snowfall measured on 2014-10-19. The spread of the data is considerable, and fitting to a power law of the form $v = cD^b$ yields $v = 0.55\,\mathrm{m\,s}^{-1} \cdot D^{-0.019}$ ($D$ in µm) with $R^2 = 0.0004$, i.e. no dependence of speed on size is found, indicated by the exponent $b$ and $R^2$ being close to

zero. The parameter $c$ coincides with the average fall speed of $0.55\,\mathrm{m\,s}^{-1}$. As mentioned in Sect. 4.3, the dominant shapes on that day were graupel and rimed needles. Using the subsets of the data representing these two shapes, now fits to the power law reveal significant correlations for graupel. However, for rimed needles the power law does not fit the data well. The results from these fits are

graupel: $v = 0.0013\,\mathrm{m\,s}^{-1} \cdot D^{0.98}$,   $D$ in µm (with $R^2 = 0.83$),

rimed needles: $v = 0.020\,\mathrm{m\,s}^{-1} \cdot D^{0.41}$,   $D$ in µm (with $R^2 = 0.054$).

These fitted power laws are shown in Fig. 11 as solid lines. The figure also shows the fall speed measured for the drizzle droplets. As expected, the droplets have the strongest dependence on size. With increasing complexity of particle shape, from droplets over graupel to rimed needles, the size dependence becomes weaker, the spread in data more, speed (at same size) slower, and $R^2$ of the fit to a power law smaller. Droplets have the simplest shape (spherical), and also the largest area ratio of

larger than 0.9. The compact graupel particles that have been selected to fit the fall speed–size relationship have a somewhat lower area ratio of on average 0.63 (with standard deviation of 0.08). Rimed needles have the lowest area ratio of on average 0.17 (st. dev. 0.04). Thus, one can also observe that with decreasing area ratio the size dependence of fall speed becomes weaker and at the same time the fit to the power law worse. So, if instead of compact graupel all particles with area ratios between 0.25 and 0.9 are selected, a group that includes graupel with more structure and smaller area ratio compared to compact graupel,

then we expect the fit quality to deteriorate. And in fact, for this group with average area ratio of 0.56 (st. dev. 0.15) the results of a fit are $v = 0.0079\,\mathrm{m\,s}^{-1} \cdot D^{0.66}$,   $D$ in $\mu$m (with $R^2 = 0.20$).

### 5   Summary

We have described the Dual Ice Crystal Imager (D-ICI), a ground-based in-situ instrument to determine snow ice crystal properties and fall speed simultaneously. Dual images are taken of detected snow particles using two CCD cameras that image

along a horizontal and close-to-vertical viewing direction, respectively. The horizontal, or side-view, is used to determine fall speed from images taken with double exposures. The close-to vertical, or top view, is used to determine size and area.

Both cameras use the same pixel resolution of approximately 4 µm/pixel. The high-resolution images provide enough detail to determine shape. The two views of the same particle avoid ambiguities in shape determination. Hence, D-ICI can be used for classification studies (Vázquez-Martín et al., 2019). Microphysical properties may then be studied specifically for certain

shapes. The necessity to discriminate shapes has been demonstrated by fitting one common power law for area versus size



to all data during a certain measurement period. The relationship that has been found would fit poorly the area for any of the shapes encountered in that period, droplets, graupel, and rimed needles. By selecting subsets of the data corresponding to certain shapes better fitting relationships have been found and reported (see Sect. 4.3). Similarly, fall speed–size relationships have been found to differ from shape to shape with varying correlations, which, however, are all better than correlation if shape

is not considered (see Sect. 4.4). Thus, an instrument that allows to discern shape and measures fall speed at the same time is required.

Snow particles fall some distance vertically through the sampling tube before images are taken, form which speed is derived. Therefore, the fall speed measurements of D-ICI are not affected by the vertical component of the wind speed or by turbulences close to ground. The accuracy of fall speed measurements has been discussed. It is limited by tumbling of snow particles.

However, tumbling is not observed frequently. Rejecting particles that tumble with a rotation of more than $10°$ as detected on the side-view image, the error can be limited to $7\%$.

Snow particle size and area are determined from top-view images, i.e. as projected along the vertical fall direction. These properties are more appropriate than the same properties determined from a horizontal view, as done by most instruments, when studying relationships to the fall speed or comparing to vertically pointing remote sensing measurements.

*Data availability.* The presented data are available at the Swedish National Data Service (DOI WILL BE ADDED AS SOON AS AVAIL-ABLE, the uploaded data is currently being evaluated by Swedish National Data Service).

*Author contributions.* TK designed and built D-ICI. TK prepared the analysis methods with contributions from SVM. TK carried out the measurements 2014–2016, SVM from 2017 with contributions from TK. TK analysed data from 2014. SVM analysed data used to evaluate frequency and extent of tumbling. TK prepared the manuscript with contributions in text and figures from SVM.

*Competing interests.* There are no competing interests.

*Acknowledgements.* We thank the Kempe Foundations (Kempestiftelserna, SMK-1024) for financial support of hardware, the Swedish National Space Agency (SNSA) for funding during the prototype development (Grant Dnr 85/10), and the Graduate School of Space Technology at Luleå University of Technology for additional financial support.



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





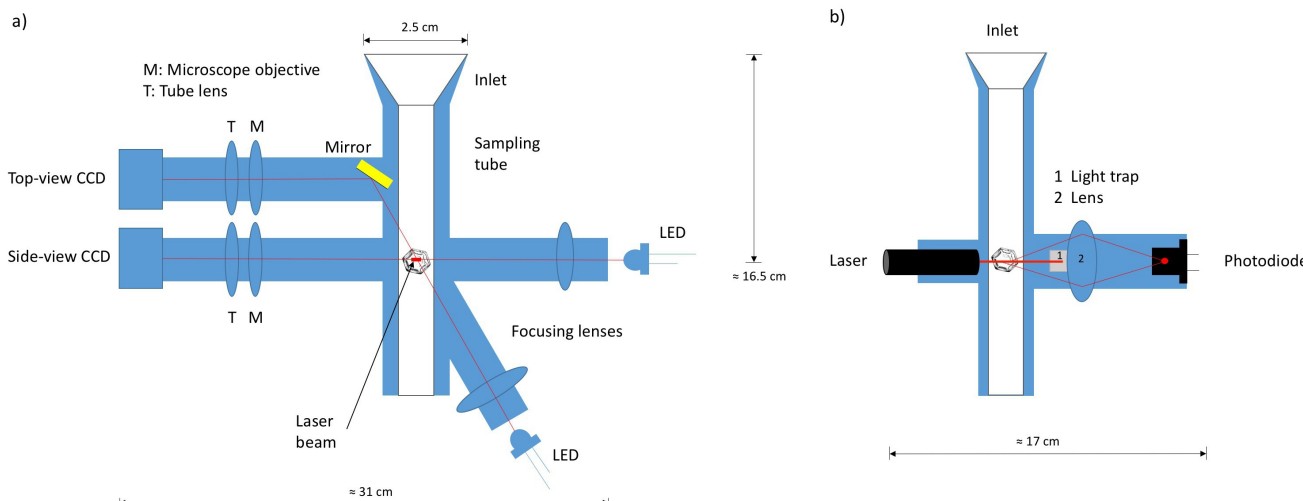

**Figure 1.** Schematic cut-views of the set-up of D-ICI. Panel a): cut through a plane defined by the optical axes of the imaging optics showing inlet, sampling tube, and the side- and top-view imaging optics and illumination; panel b): perpendicular cut showing laser detection consisting in laser, light trap, lens for collection of scattered light, and photodiode. In both panels the optical cell with the sensing volume at its centre is indicated by the image of an ice crystal (not to scale).

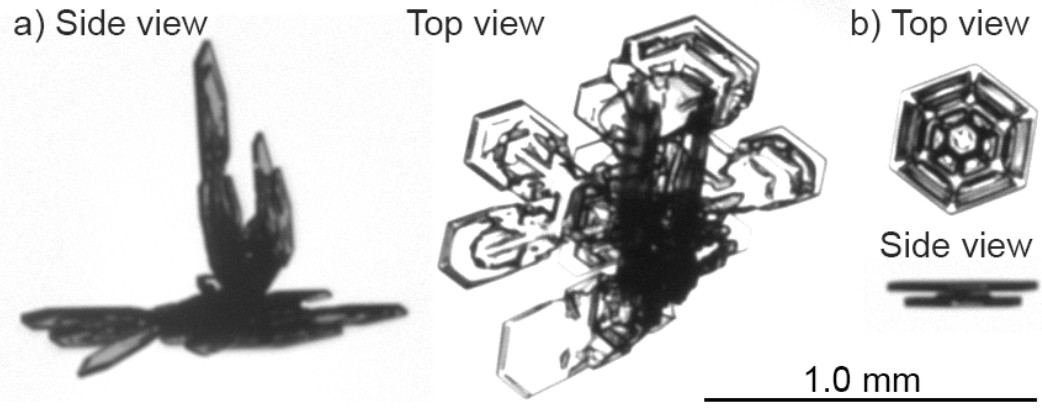

**Figure 2.** Two examples of ice crystals imaged in two viewing geometries, top view and side view. The ice crystal shown in panel a) has a width of approximately 1.2 mm, the one in panel b) 0.4 mm. Both ice crystals in panel a) and b) use the same scaling, and, for reference, a size bar with length corresponding to 1 mm (and width of 10 μm) is shown.



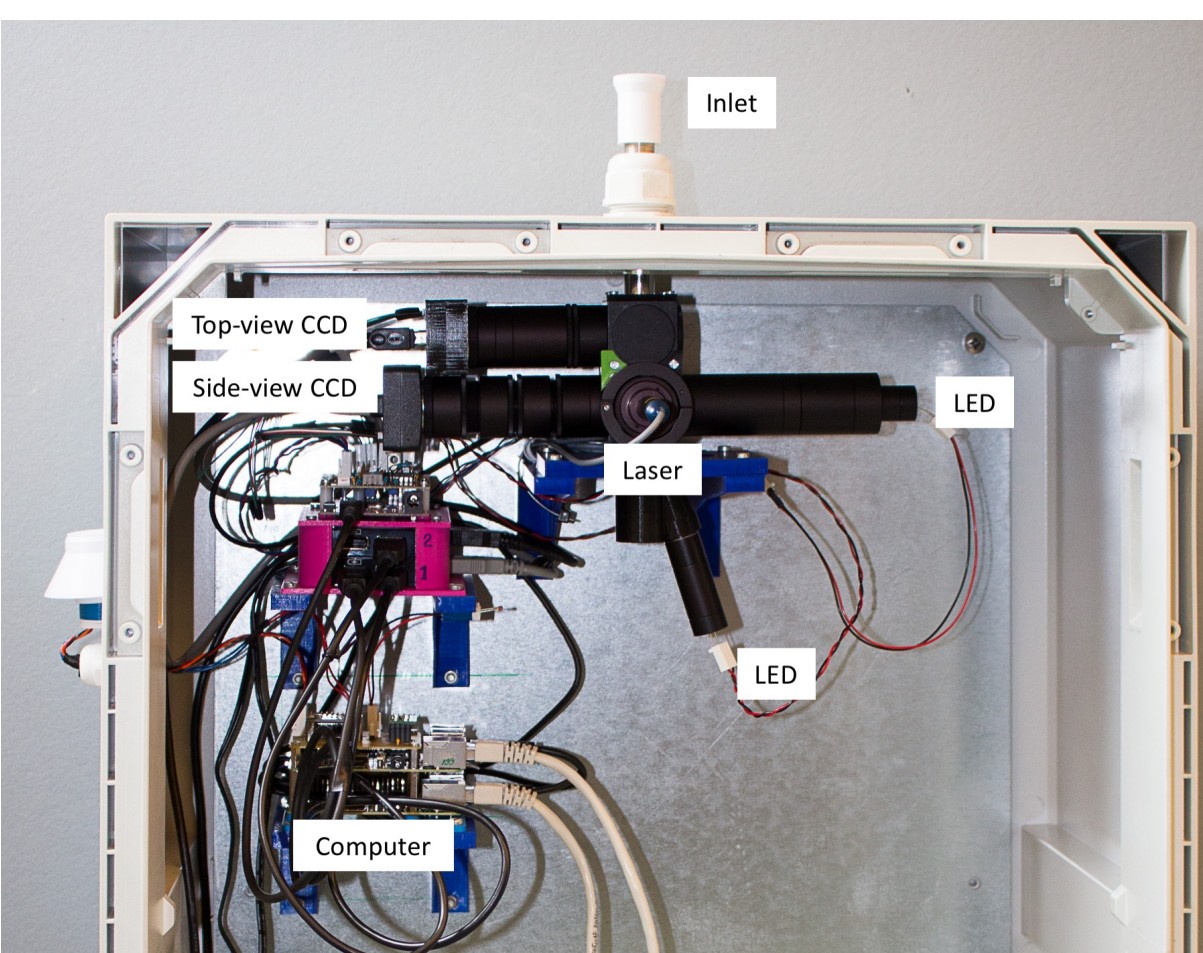

**Figure 3.** Photograph of D-ICI (door of enclosure is removed).



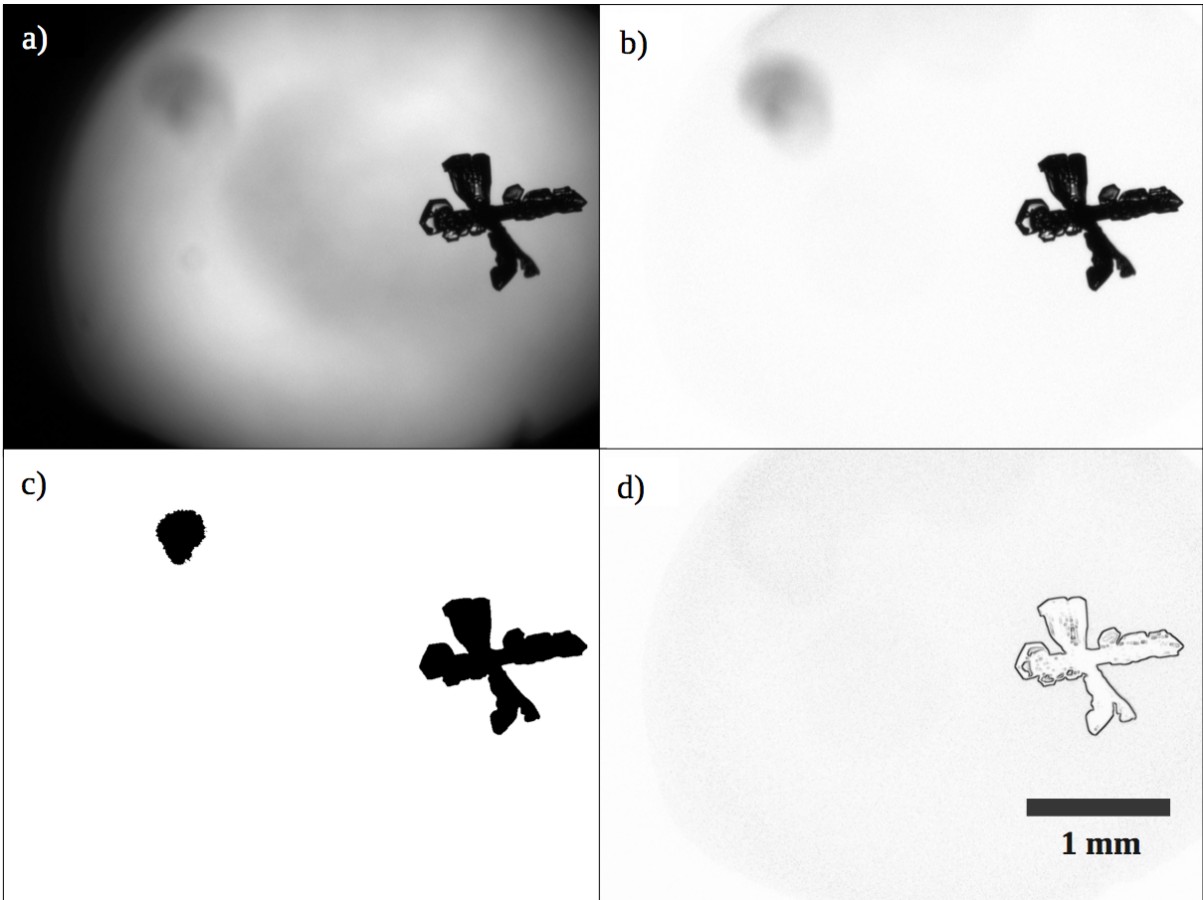

**Figure 4.** Automated image processing steps shown for an example image. Panel a) shows the original image; b) cleaned image (background features removed); c) binary mask, where logical True values correspond to regions on the cleaned image that are darker than the grey-level threshold, here shown as black; d) gradient matrix computed from the cleaned image, values scaled to grey levels for representation (largest gradient value corresponds to black and zero gradient to white). See description in text for details of the processing procedure. The resolution is indicated by a size bar of 1 mm.





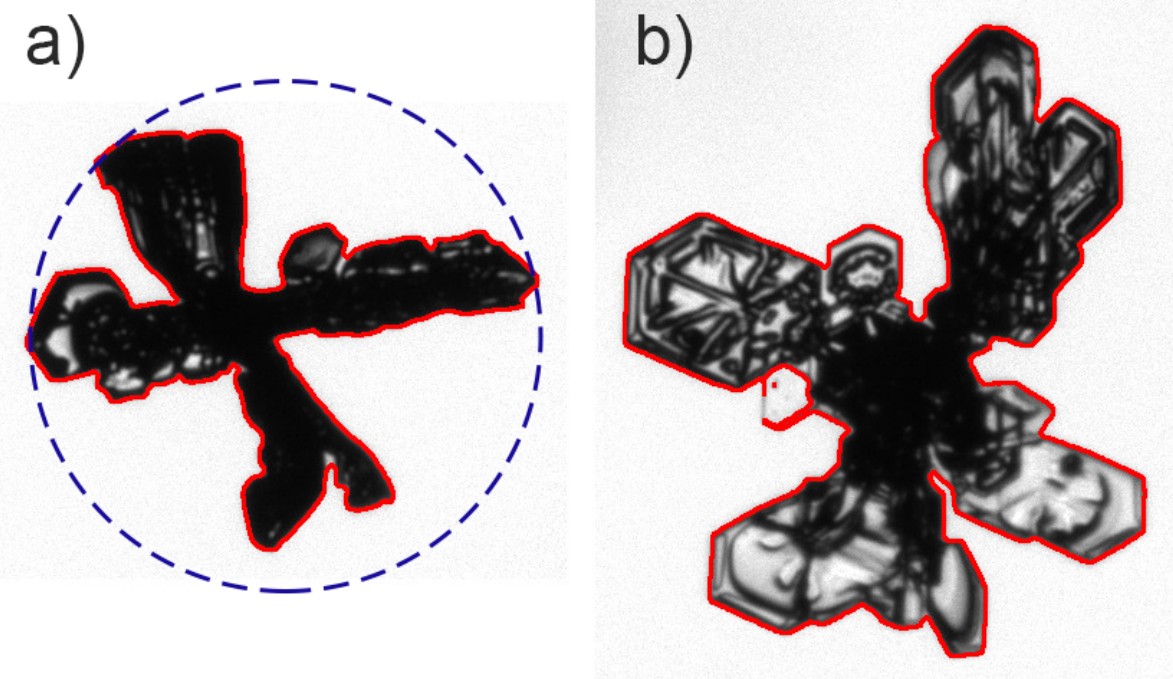

**Figure 5.** Detected edges of processed ice particle images. The edges are shown in red and have been enlarged to a thickness of 3 pixel for better visibility in this figure. One example, panel a), shows the edge of the ice particle from Fig. 4. The smallest circle enclosing the particle is shown with a dashed line; its diameter, i.e. the maximum dimension of the ice particle, is 1.34 mm (or 358 pixel). The other example in panel b) shows an ice particle that has been detected with fragmented edge due to parts of the actual particle edge being too bright (see text for more details).

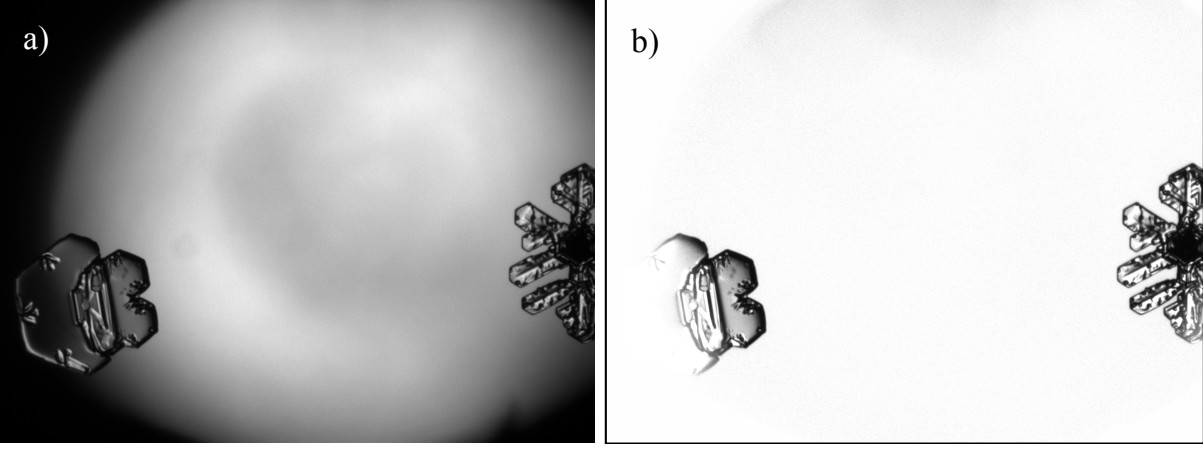

**Figure 6.** Examples of ice particles flagged as 'on-border' (right) and 'in-darkregion' (left). The original image a) and the image b) after background removal are shown.

**Figure 7.** Ice particles as imaged in two viewing geometries, top view and side view. Each ice particle is shown as a pair of these two views, with the top view in the upper panel, and the corresponding side view in the respective lower panel. Two rows of such pairs are shown. All images have the same resolution, for reference a size bar with length corresponding to 1 mm is shown.





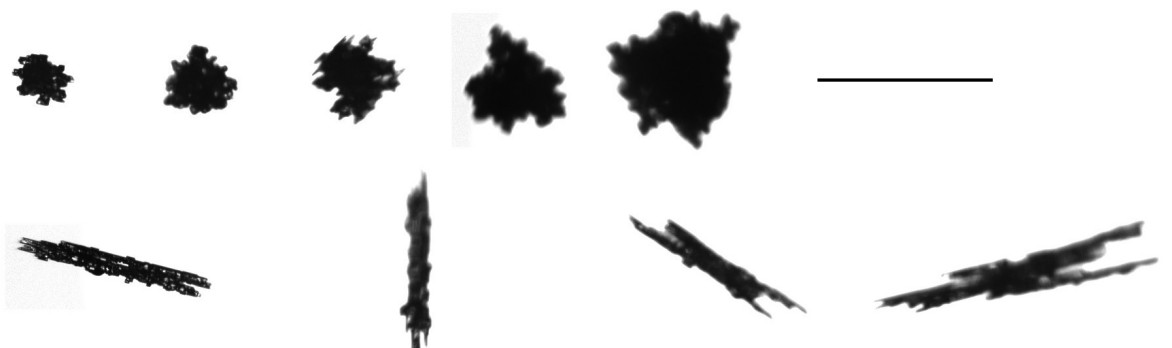

**Figure 8.** Example images of the two shapes, graupel (top row) and rimed needles (bottom row), from the snowfall measured on 2014-10-19.
For reference a size bar with length corresponding to 1 mm is shown.

**Figure 9.** Example side-view images of doubly-exposed falling ice particles. The fall speed is determined from the vertical separation of the two instances of the particle on the same image. The top row shows measurements from 2014-10-19, the bottom row from 2014-10-23. For reference a size bar with length corresponding to 1 mm is shown.

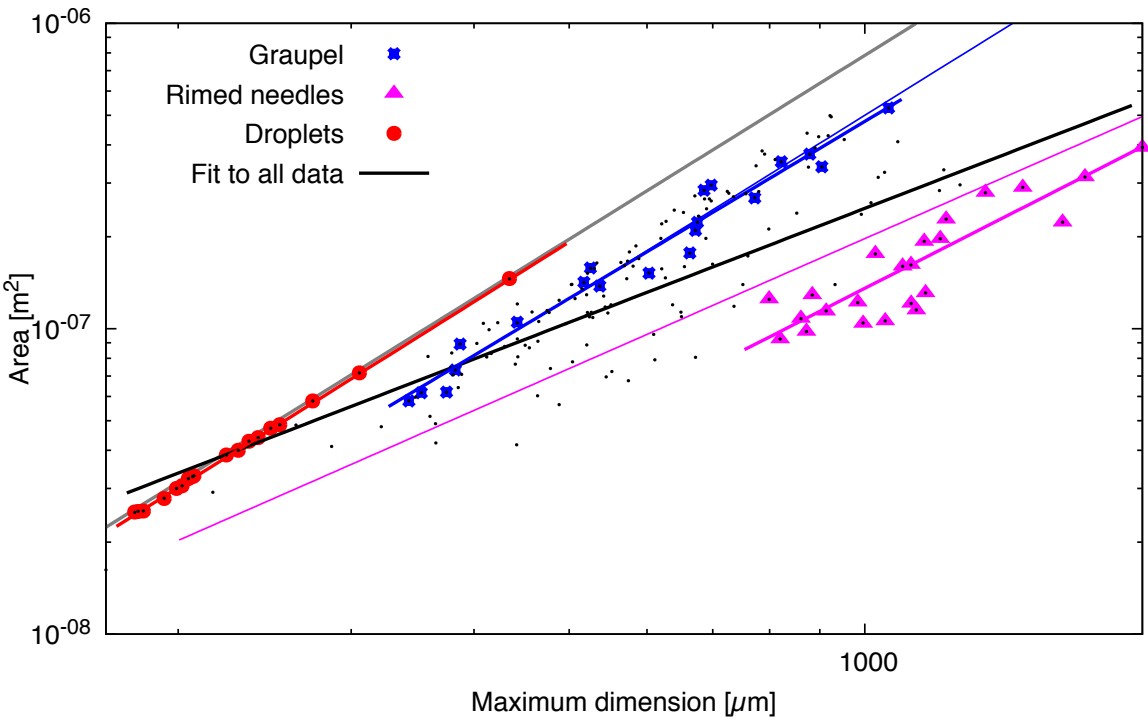

**Figure 10.** Area $A$ versus maximum dimension $D$ from snowfall measured on 2014-10-19 between approximately 6 and 19 UTC. Fits to all data (black dots) and to three subsets corresponding to graupel, rimed needles, and droplets are shown as lines in the same colour as the corresponding data points. For comparison, a grey line indicates the cross-sectional area of spheres. In addition, two relationships reported by Mitchell (1996) are shown as thinner lines, one for rimed long columns (magenta) and one for lump graupel (blue).



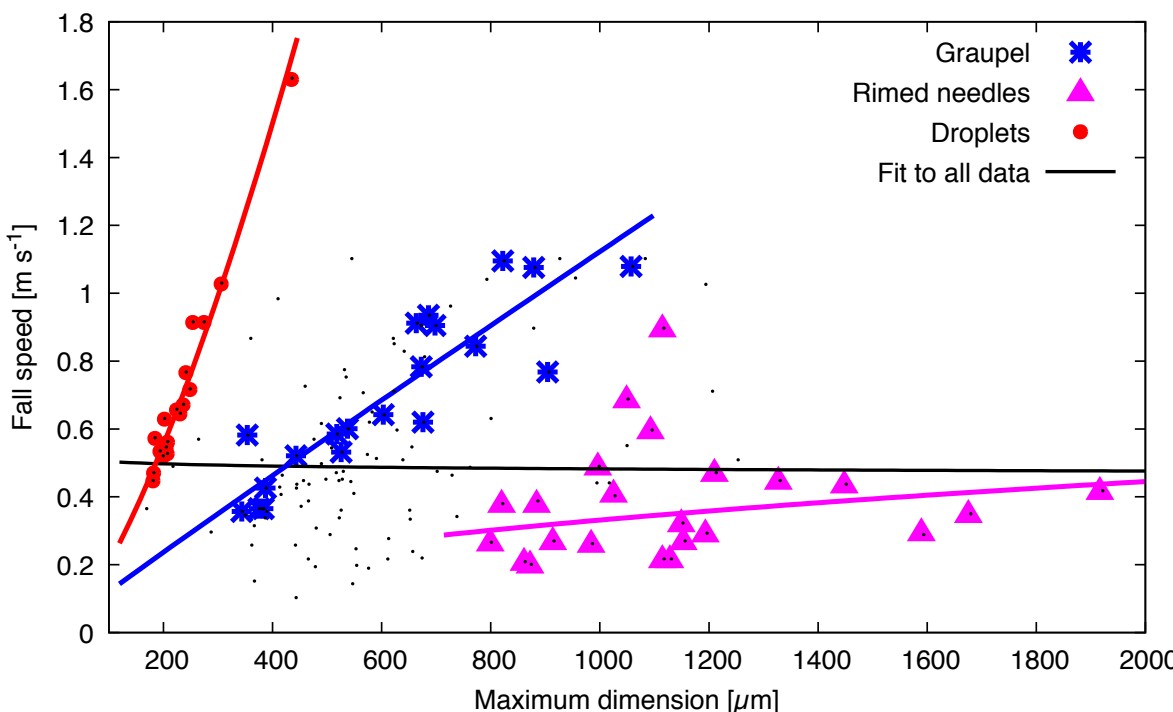

**Figure 11.** Fall speed versus maximum dimension $D$ for snowfall measured on 2014-10-19 between approximately 6 and 19 UTC. Fits to all data (black dots) and to three subsets corresponding to graupel, rimed needles, and droplets are shown as lines in the same colour as the corresponding data points.