# Peer review of "Microphysical properties and fall speed measurements of snow ice crystals using the Dual Ice Crystal Imager (D-ICI)"

_Atmospheric Measurement Techniques, 2019_

## Referee Comment (RC1) · Kevin Hammonds (Referee) · 3 Dec 2019

I have reviewed the manuscript, "Microphysical properties and fall speed measurements of snow ice crystals using the Dual Ice Crystal Imager (D-ICI)", submitted for publication to Atmospheric Measurement Techniques. In this manuscript, the authors present a detailed description, methodology, and application for a newly developed high-resolution ice particle imager that can be used to characterize ice crystal habits and fall speeds of these frozen hydrometeors without the influence of ground-level wind or turbulence. Overall, I found the manuscript to be very well-written and organized, with only minor typographic or grammatical errors, and I found all figures appropriately

and adequately represented. Additionally, the subject matter fits well within the scope of the selected publication.

Although there are currently several other similar instruments that have been developed in recent years for similar applications, as summarized in the manuscript, I found the D-ICI to be novel in that it appears to be the first of such instruments to incorporate a downward-looking viewing angle, such that two orthogonal planes of the ice crystal geometry are captured (i.e. parallel and perpendicular to the falling direction) when the ice crystal falls through the viewing area, allowing for a more accurate estimate of the maximum dimension of the particle. As an example of the effectiveness of their approach and instrumentation, the authors go on to show comparisons to well-known literature on the topic of classifying frozen hydrometeor type via power law relationships relating the particle area to its maximum dimension of (e.g. Mitchell 1996), with which their results compared quite well. Furthermore, the authors demonstrate from field data, the relevance in identifying ice crystal type as a means for correlating fall speed with the maximum dimension, such that maximum dimension alone cannot be used to predict the fall speed of such particles.

In conclusion, I congratulate the authors on their efforts and recommend the manuscript for publication upon completing only Minor Revisions.

**Detailed Comments:**

Line 23: delete the letter "s" in the word "remotes"

Lines 25-27: Citation for linking snowfall rate to snow depth on the ground?

Line 26: add word "role" after "significant"

Line 29: additional citations to Sun et al 2011 that are of relevance to the topic and may be of interest to readers:

Matrosov, S. Y., Mace, G. G., Marchand, R., Shupe, M. D., Hallar, A. G., & McCubbin, I. B. (2012). Observations of ice crystal habits with a scanning polarimetric W-band
radar at slant linear depolarization ratio mode. Journal of Atmospheric and Oceanic Technology, 29(8), 989-1008.

Marchand, R., Mace, G. G., Hallar, A. G., McCubbin, I. B., Matrosov, S. Y., & Shupe, M. D. (2013). Enhanced radar backscattering due to oriented ice particles at 95 GHz during StormVEx. Journal of Atmospheric and Oceanic Technology, 30(10), 2336-2351.

Line 182-183: Citation for assumed fall speed?

Line 188-189: Can you comment on any uncertainty associated with these estimates?

Line 201: add the letter "a" after "appear as"

Line 252: add letter "n" to "know,"

- Line 256: add letter "a" after "As"
- Line 304: add the word "the" after "when"

Line 387: Replace "form" with "from"

---

## Referee Comment (RC2) · Timothy Garrett (Referee) · 17 Dec 2019

Overall I find this paper well worth publication in AMT for introducing a new instrument that can be applied to the concurrent fallspeed and imaging of small precipitation particles. It is an advance over widely used instrumentation that provides similar quality images but no direct measurement of fallspeed. The D-ICI instrument and data processing are thoroughly described and initial results are provided the are largely consistent with expectations.

I have only a few substantial comments.

[Figure]

1. The writing is rather idiosyncratic at times and could use a professional edit

2. Section 2 about the inlet and sampling tube is insufficiently supported. The inlet design is such that in sufficiently high winds I could easily imagine based on experience and prior literature such effects as poor sampling, induced tumbling, crystal fracturing, and altered particle fall speeds. The paper states currently "The length of the sampling tube upstream of the sensing volume is sufficient (more than ten times the diameter of the sampling tube) so that particles can relax from any effects of wind. Hence, the fall speed of ice particles is not affected by wind or turbulence," but without justification that would lend real confidence.

There is an extensive literature on particle sampling by inlets, even in the atmospheric sciences, back-of-the-envelope calculations could be done, and Computational Fluid Dynamics simulations can also be performed relatively easily in e.g. CAD. I feel that some improvement is needed here.

3. A limitation of the device that should be acknowledged for particle classification is that the larger particles are near silhouettes. I would say that the rounded particles in the top row of Figure 8 could just as easily be assemblages of small crystals as graupel, particularly given their more structured boundaries.

4. Figure 10 includes prior results by Mitchell. What not show the same comparison in Figure 11? There are many possible sources, e.g. Locatelli and Hobbs.

---

## Author Comment (AC1) · 4 Feb 2020

The response is attached as supplement. It is compiled into one PDF containing the Response to both RC1 and RC2 as well as the revised manuscript in a version with highlighted changes.

Please also note the supplement to this comment: https://www.atmos-meas-tech-discuss.net/amt-2019-352/amt-2019-352-AC1-supplement.pdf

---

## Author Response (AR1)

**Final response to reviewers' comments — amt-2019-352**

Dear Editor and Reviewers,

Thank you for taking the time to assess our article amt-2019-352. Below are our point-by-point answers (in blue) to the suggestions and questions posed by the reviewers (in black).

Additionally, a few further changes to the manuscript are listed. Together with these answers, I am also sending a version of the revised manuscript with highlighted changes.

Yours sincerely,

Thomas Kuhn and co-author
* * *
Point-by-point response to detailed comments: RC: Referee's comment, AR: authors' response, ACM: authors' changes in manuscript.

**Author Comment in response to Referee Comment amt-2019-352-RC1 by Kevin Hammonds (Referee):**

I have reviewed the manuscript, "Microphysical properties and fall speed measurements of snow ice crystals using the Dual Ice Crystal Imager (D-ICI)", submitted for publication to Atmospheric

Measurement Techniques. In this manuscript, the authors present a detailed description, methodology, and application for a newly developed high-resolution ice particle imager that can be used to characterize ice crystal habits and fall speeds of these frozen hydrometeors without the influence of ground-level wind or turbulence. Overall, I found the manuscript to be very well-written and organized, with only minor typographic or grammatical errors, and I found all figures appropriately and adequately represented. Additionally, the subject matter fits well within the scope of the selected publication.

Although there are currently several other similar instruments that have been developed in recent years for similar applications, as summarized in the manuscript, I found the D-ICI to be novel in that it appears to be the first of such instruments to incorporate a downward-looking viewing angle, such that two orthogonal planes of the ice crystal geometry are captured (i.e. parallel and perpendicular to the falling direction) when the ice crystal falls through the viewing area, allowing for a more accurate estimate of the maximum dimension of the particle. As an example of the effectiveness of their approach and instrumentation, the authors go on to show comparisons to well-known literature on the topic of classifying frozen hydrometeor type via power law relationships relating the particle area to its maximum dimension of (e.g. Mitchell 1996), with which their results compared quite well. Furthermore, the authors demonstrate from field data, the relevance in identifying ice crystal type as a means for correlating fall speed with the maximum dimension, such that maximum dimension alone cannot be used to predict the fall speed of such particles.

In conclusion, I congratulate the authors on their efforts and recommend the manuscript for publication upon completing only Minor Revisions.

Detailed Comments:

RC1-1) Line 23: delete the letter "s" in the word "remotes"

AR: Thank you for spotting this mistake.

ACM: The error has been corrected.

RC1-2) Lines 25–27: Citation for linking snowfall rate to snow depth on the ground?

AR: To predict precipitation amount reaching the ground, one needs to know microphysical properties of the snow particles that are falling down. This statement is relevant to our study. While snow depth is certainly important too, it cannot be retrieved from the microphysical properties of the falling snow, but is rather a consequence of the precipitation amount and several other factors such as temperature on the ground after the snowfall. As such, our statement in Lines 25–27 was wrong, and a reference to a study that links snowfall rate directly to snow depth is not relevant here.

ACM: We are not mentioning 'snow depth' in the revised manuscript.

RC1-3) Line 26: add word "role" after "significant"

AR: Thank you for catching this mistake.

ACM: The error has been corrected.

RC1-4) Line 29: additional citations to Sun et al 2011 that are of relevance to the topic and may be of interest to readers:

Matrosov, S. Y., Mace, G. G., Marchand, R., Shupe, M. D., Hallar, A. G., & McCubbin, I. B. (2012). Observations of ice crystal habits with a scanning polarimetric W-band radar at slant linear depolarization ratio mode. Journal of Atmospheric and Oceanic Technology, 29(8), 989–1008.

Marchand, R., Mace, G. G., Hallar, A. G., McCubbin, I. B., Matrosov, S. Y., & Shupe, M. D. (2013). Enhanced radar backscattering due to oriented ice particles at 95 GHz during StormVEx. Journal of Atmospheric and Oceanic Technology, 30(10), 2336–2351.

AR: Thank you for the suggestion.

ACM: References have been added.

RC1-5) Line 182–183: Citation for assumed fall speed?

AR: To determine the sampling volume $V$ we have assumed a fixed fall speed $v$ of $0.5\,\mathrm{m\,s^{-1}}$. This assumption was based on our data presented in this study, which had an average fall speed of approx. $0.5\,\mathrm{m\,s^{-1}}$.

ACM: As this was not clear, we have now mentioned why we have assumed $0.5\,\mathrm{m\,s^{-1}}$: "To determine $V$ a constant fall speed $v$ of $0.5\,\mathrm{m\,s^{-1}}$ is assumed, which corresponds approximately to the average fall speed of the data used here."

RC1-6) Line 188–189: Can you comment on any uncertainty associated with these estimates?

AR: The size dependencies of the sensing area and the probability of the particle being partially outside the FOV cancel out to a good approximation (see Sect. 2.4). This size dependency may be corrected, and the correction factor for number concentration would vary between 1.07 and 1.09 for particles with maximum dimensions between 1.0 and 2.0 mm, reaching down to a minimum of 1.03 for particles of 1.4 mm. And for particles down to 0.5 or up to 2.5 mm it would increase to approximately 1.25. The assumption of constant snow fall speed $v$, mentioned above, introduces a new uncertainty. When the constant speed of $0.5\,\mathrm{m\,s^{-1}}$ is overestimating the actual particle fall speed, then the concentration $n$ of these particles is underestimated. And conversely, underestimating the speed results in overestimating concentration. About two third of the data used here had fall speeds between 0.3 and $0.85\,\mathrm{m\,s^{-1}}$, that means that for those particles the error in concentration ranges from, respectively, underestimating concentration by about 40% to overestimating it by 70%. This may be corrected for with correction factors based on measured fall speed, which would then vary for the two third of data considered here between about 1.7 and 0.6, respectively. An additional uncertainty in estimating the effective sensing area is resulting from the uncertainty in determining the laser beam width, which may be on the order of $\pm 20\%$, however is difficult to measure.

These uncertainties affect both $n$ and $r_\mathrm{s}$. We have not yet verified the uncertainties experimentally. Hence, $n$ and $r_\mathrm{s}$ determined with D-ICI using the assumptions and estimates outlined above should be considered estimates of the actual number concentration and snowfall rate.

ACM: These additional comments on the uncertainties may be useful in the revised manuscript. Therefore the above explanations are replacing the paragraph in question.

RC1-7) Line 201: add the letter "a" after "appear as"

AR: Thank you.

ACM: The error has been corrected.

RC1-8) Line 252: add letter "n" to "know,"

AR: Thank you.

ACM: The error has been corrected.

RC1-9) Line 256: add letter "a" after "As"

AR: Thank you.

ACM: The error has been corrected.

RC1-10) Line 304: add the word "the" after "when"

AR: Thank you.

ACM: The error has been corrected.

RC1-11) Line 387: Replace "form" with "from"

AR: Thank you.

ACM: The error has been corrected.

**Author Comment in response to Referee Comment amt-2019-352-RC2 by Timothy Garrett (Referee):**

Overall I find this paper well worth publication in AMT for introducing a new instrument that can be applied to the concurrent fallspeed and imaging of small precipitation particles. It is an advance over widely used instrumentation that provides similar quality images but no direct measurement of fallspeed. The D-ICI instrument and data processing are thoroughly described and initial results are provided the are largely consistent with expectations.

I have only a few substantial comments.

RC2-1) The writing is rather idiosyncratic at times and could use a professional edit.

AR: We have modified some sections of the paper, so the writing has been improved according to your suggestion.

ACM: Mainly Sect. 1 has been improved.

RC2-2) Section 2 about the inlet and sampling tube is insufficiently supported. The inlet design is such that in sufficiently high winds I could easily imagine based on experience and prior literature such effects as poor sampling, induced tumbling, crystal fracturing, and altered particle fall speeds. The paper states currently "The length of the sampling tube upstream of the sensing volume is sufficient (more than ten times the diameter of the sampling tube) so that particles can relax from any effects of wind. Hence, the fall speed of ice particles is not affected by wind or turbulence," but without justification that would lend real confidence.

There is an extensive literature on particle sampling by inlets, even in the atmospheric sciences, back-of-the-envelope calculations could be done, and Computational Fluid Dynamics simulations can also be performed relatively easily in e.g. CAD. I feel that some improvement is needed here.

AR: We agree that wind speed is important to take into account since it affects snow measurements. In the future, design improvements and/or quantifying such effects of wind on our snow measurements should be considered. However, in this paper we have focused on the instrument description in its current configuration. A particular focus has been on the dual imaging and size, area, and shape determination, which are not affected by wind. The study case presented in this paper, data from 2014-10-19, does not have strong winds, and the average of the wind speed was $2\,\mathrm{m\,s^{-1}}$ (`https://www.smhi.se/data/meteorologi/ladda-ner-meteorologiska-observationer/#param=wind,stations=all,stationid=180940`). It is therefore suitable for demonstrating typical measurements.

ACM: We have highlighted the limitations with respect to high wind speeds. We have modified accordingly the two sentences that you have cited in Sect. 2.2. In addition, we

have included wind effects in the discussion of uncertainties in Sect. 3.1. We have also stated in Sect. 4.2 that the data considered do not include high wind speeds.

RC2-3) A limitation of the device that should be acknowledged for particle classification is that the larger particles are near silhouettes. I would say that the rounded particles in the top row of Figure 8 could just as easily be assemblages of small crystals as graupel, particularly given their more structured boundaries.

AR: The set-up with background illumination enables one to see details in the structure of relatively simple snow crystals. For more complex snow crystals or aggregates, many details are hidden due to larger parts of the snow particle becoming opaque for the illumination. This can be considered a limitation of the illumination scheme and it is not always possible to unambiguously determine the shape. However, in many cases that seem ambiguous as judged from one image only, classification is possible due to the second image from a different point of view. For example, a needle or a stellar crystal from the side may be largely opaque, whereas seen from the top the stellar crystal would reveal more detail (see Fig. 1). While some ambiguities may remain in a few cases of largely opaque snow particles, in many cases some structure that can be seen close to the edge of the silhouette can provide hints on the shape or morphology of the snow particle. Having two images of the same particle also helps in these cases, as these two views will increase the possibility to see such helpful structural details somewhere on the particle. Fig. 2 shows two examples, one snow particle with riming and one without. While it seems safe to us to assume that the particle in panel a) features riming, it is not clear if the riming happened on a single ice crystal or on an agglomerate. For comparison, panel b) shows an agglomerate that appears to be without riming.

The rimed snow particles from Fig. 8 in the manuscript are shown again in Fig. 3, with

the same reference for size, a 1 mm long bar. The top row shows the top view (i.e. the same images as in Fig. 8 of the manuscript); the bottom row shows the corresponding side views. If considering both views, then one can make out small features at the particle silhouette that could originate from small droplets that attached during riming to the particle (small roundish features with a bright spot in the centre, typical for droplets, with a diameter of around 6 pixels or 23 µm).

ACM: The limitation of the imaging set-up has been mentioned in Sect. 2.3. In Sect. 5, the advantage of having two images from two different viewing directions for shape classification has been mentioned more clearly, while at the same time mentioning the limitations.

RC2-4) Figure 10 includes prior results by Mitchell. What not show the same comparison in Figure 11? There are many possible sources, e.g. Locatelli and Hobbs.

AR: This a good suggestion. The cited study by Mitchell (1996) reports many relationships of previous studies, the 'Lump graupel' we are showing, for example, is from Locatelli and Hobbs (1974). Mitchell (1996) has used these relationships to determine fall speed for the different corresponding shape. Therefore, we are showing in Fig. 11 the two fall speed relationships determined for lump graupel and rimed long columns, for which we had included the area relationships for comparison in Fig. 10.

ACM: We have added the fall speed relationships determined from Mitchell (1996). The text in Sect. 4.4, where Fig. 11 is explained, mentions now the comparison with these fall speed relationships. The legends in Figures 10 and 11 have been updated to include the previous studies shown for comparison.

[Figure]

Figure 1: A needle or a stellar crystal from the side may be largely opaque, whereas seen from the top the stellar crystal would reveal more detail allowing correct classification. Panel a): stellar crystal, left shows top and right shows side view (exposed twice); panel b): needle, left shows top and right shows side view (exposed twice).

[Figure]

Figure 2: Panel a): a particle that shows signs of riming; and panel b): an agglomerate of single crystals with no or very little riming. The snow particle in panel a) shows features of riming close to its silhouette in addition to few details of the underlying crystal structure. While the riming seems clear, it remains unclear if the underlying ice particle is a single crystal or an agglomerate.

[Figure]

Figure 3: The top row shows the top view of the rimed particles of Fig. 8 in the manuscript. The bottom row shows the corresponding side views. If considering both views, then one can make out small features at the particle silhouette, which could originate from small droplets that attached during riming to the particle (small roundish features with a bright spot in the centre, typical for droplets, with a diameter of around 6 pixels or 23 μm).

**References**

Locatelli, J. D. and Hobbs, P. V.: Fall speeds and masses of solid precipitation particles, J. Geophys. Res., 79, 2185–2197, 1974.

[revised manuscript text omitted]